# Transcriptome Analysis Reveals Genetic Factors Related to Callus Induction in Barley

Zhengyuan Xu [1], Fengyue Wang [1], Yishan Tu [1], Yunfeng Xu [1], Qiufang Shen [1,2,*] and Guoping Zhang [1,2]

1 Department of Agronomy, Key Laboratory of Crop Germplasm Resource of Zhejiang Province, Zhejiang University, Hangzhou 310058, China; 11816026@zju.edu.cn (Z.X.); 22116018@zju.edu.cn (F.W.); 21916141@zju.edu.cn (Y.T.); 12016014@zju.edu.cn (Y.X.); zhanggp@zju.edu.cn (G.Z.)
2 Zhongyuan Institute, Zhejiang University, Zhengzhou 450000, China
* Correspondence: shenqf@zju.edu.cn

**Abstract:** Barley is an important cereal crop worldwide. Its genetic transformation is now limited to very few cultivars because of the high genotype dependence of embryogenic callus. To reveal the key genes or factors controlling the callus induction and plantlet regeneration in barley, we compared the transcriptomic profiles of immature embryos of Golden Promise and ZU9, which differed dramatically in the efficiency of the genetic transformation. The samples were taken at 0, 5, 10 and 20 days of the culture, respectively. In total, 5386 up-regulated and 6257 down-regulated genes were identified in Golden Promise. Several genes, identified exclusively in GP callus, were selected for further investigation. These genes were mainly involved in protein metabolism, energy metabolism, stress response, detoxification and ubiquitin–proteasome. Four *YUCCA flavin monooxygenases*, one *PIN-FORMED*, one *tryptophan aminotransferase related*, three *small auxin up RNA*, three *indole-3-acetic acid* and one *adenylate isopentenyl transferase*, seven *cytokinin oxidase/dehydrogenase*, three *Arabidopsis histidine kinase*, three *Arabidopsis histidine phosphotransfer protein*, and one *Arabidopsis response regulator* were differentially expressed in the calli of the two barley genotypes, suggesting that biosynthesis, response and transport of auxin and cytokinin might be associated with cell reprogramming during callus induction. The current results provide insights into molecular mechanisms of callus induction at an early developmental stage and are helpful for optimizing the tissue culture system in barley.

**Keywords:** barley; callus induction; transcriptome; genetic difference; auxin signaling pathway



## 1. Introduction

Plant tissue culture plays an essential role in genetic transformation for functional research of genes, and also provides a feasible and promising approach to produce transgenic plants. Currently, *Agrobacterium*-mediated gene transfer is the most effective in genetic transformation [1]. In modern breeding, once the elite genotype with favorable agronomic traits is determined, the transfer of the target gene to the genotype can be performed [2,3]. For the successful transfer of the objective gene, development of the efficient tissue culture system is necessary [4].

There are many factors affecting the efficiency or success of the plant tissue culture, while plant genotype is a determinative one [5–7]. In rice (*Oryza sativa* L.), the genetic transformation of *japonica* varieties has long been developed [8], but it is still quite difficult for genetic transformation of *indica* due to low efficiency of callus induction and plantlet regeneration [9,10]. *Japonica* varieties 'Nipponbare' and 'Kitaake', mainly used in the genetic transformation of rice, are notably easier to be transformed in comparison with *indica* varieties such as 'IR64' and 'Ciherang-Sub1' [11]. In the genetic transformation of maize (*Zea mays* L.), the difficulty in embryogenic induction and regeneration occurs for many genotypes [12], which greatly restricts the scope of genetic transformation in maize. Inbred 'A188' and 'Hi-II' are two genotypes commonly used in maize tissue culture, while other genotypes are quite low in the frequency of genetic transformation [13–15].

Obviously, the degree of difficulty in callus induction and plant regeneration varies greatly with species or genotypes [16], which are in turn genetically controlled. It was reported that eight quantitative trait loci and three epistatic interactions are involved in callus induction and formation in a maize recombinant inbred line population [17,18]. Wheat (*Triticum aestivum* L.) is recalcitrant to tissue culture due to its large size of genome. Its genetic transformation is now limited to a few cultivars such as 'Bobwhite' and 'Fielder' [19]. Apparently, the genotype dependence is a major bottleneck restricting cereal genetic transformation. Therefore, it is imperative to identify the genes or loci controlling callus induction and plant generation for overcoming the genotypic dependence of genetic transformation.

In addition to plant genotypes, successful transformation also relies on suitable medium components such as exogenous phytohormones [6]. Auxin and cytokinin, which regulate cell growth, development and differentiation, are generally exogenously applied to the medium for tissue culture [20]. Auxin signaling is transduced by auxin response factors (ARF). It was reported that ARF7 and ARF9 can enhance the expression of E2F TRANSCRIPTION FACTOR a (E2Fa) and LATERAL ORGAN BOUNDARIES DOMAIN (LBD) [21,22]. In turn, *LBD* can activate the expression of a series of genes promoting cell proliferation [23–25]. In *Arabidopsis*, overexpression of *LBD16*, *LBD17*, *LBD18* and *LBD29* facilitated callus induction [21].

Barley (*Hordeum vulgare* L.) is the fourth largest cereal crop worldwide in terms of planting areas and has multiple uses, including malting, animal feed and human food [26]. Moreover, genome sequence and pan-genome of barley were recently published [27,28]. Thus, barley is regarded as a model *Triticeae* crop for genetic research [29]. However, genetic transformation of barley remains limited due to its high genotype-specific dependence and now only immature embryos of 'Golden Promise (GP)' are used as explants [30,31]. Golden Promise, showing high regeneration capacity, has been identified as the most reliable and suitable model cultivar for stable genetic transformation in barley [32]. This situation poses a great challenge to genetic transformation of barley, as the model cultivar GP is an old cultivar and does not show desirable agronomic traits. Furthermore, little knowledge is available about molecular mechanisms underlying the difference between GP and other barley genotypes in the capacity of genetic transformation.

Therefore, it is vital to develop a genetic transformation system which is not so dependent on genotypes or perform well for any genotype with favorable agronomic traits. 'ZU9' is a new barley cultivar with high yield potential and malting quality in Zhejiang, China. In the previous study, we found that calli from ZU9 were soft and watery, and unable to regenerate. Hence, for understanding the reasons why Golden Promise is more amenable than ZU9 for genetic transformation, we compared the transcription regulation pathways between calli of GP and ZU9 at the different developmental stages to understand the major factors causing the difference in genetic transformation between barley genotypes.

## 2. Materials and Methods

### 2.1. Plant Materials and Growth Condition

Two barley genotypes, i.e., 'Golden Promise' (a conventional cultivar widely-used in genetic transformation) and 'ZU9' (a malting cultivar being planted in Zhejiang, China), were grown in a plant chamber with a 16 h/8 h of light/dark, 18–22 °C and 70% of relative humidity. Immature seeds were collected from barley spikes at 2–3 weeks after anthesis.

### 2.2. Sampling

Immature embryos were obtained according to Harwood et al. [33]. with some modification. Immature barley seeds were sterilized with 70% (*v/v*) ethanol for 30 s and rinsed 3 times with sterile distilled water. After that, they were immersed in 50% (*v/v*) sodium hypochlorite for 4 min and washed with sterilized water at least 4 times. Then, the scutella were isolated from the seeds by removing the embryonic axes with a pair of fine-point tweezers, and were placed on Callus Induction (CI) medium for 2–3 d. The callus along

with embryos were transferred to fresh callus induction or selection medium (CI/CIS) in dark at 23 °C every two weeks. Some calli were discarded when they turned brown, stopped growing or were contaminated. After 4–6 weeks of culture, calli were then transferred to the transition medium under low light for differentiation. Meanwhile, immature embryos or calli cultured for 0, 5, 10 and 20 days (D0, D5, D10 and D20) were collected, frozen immediately in liquid nitrogen and then stored at −70 °C. Each sample, used for RNA sequencing (RNA-Seq), consisted of about 60 immature embryos with three replicates.

### 2.3. Medium Preparation

The medium components were used according to Harwood et al. [33]. For callus induction (CI), the medium contains (per liter): 4.3 g MS salt base (Phyto Technology Laboratories, Lenexa, KS, USA), 30 g maltose, 1.0 g casein hydrolysate, 350 mg myo-inositol, 690 mg proline, 1.0 mg thiamine HCl, 2.5 mg dicamba, 1.25 mg $CuSO_4 \cdot 5H_2O$, 3.5 g phytagel. For callus induction selection (CIS), 50 mg/L hygromycin and 160 mg/L timentin were added to the CI medium, used as selective marker of target genes and antibacterial drugs, respectively. For transition (T), the medium contains (per liter): 2.5 mg 2,4-D, 0.1 mg 6-BA, 2.7 g MS-modified salt base (without $NH_4NO_3$) (Coolaber Science & Technology, Beijing, China), 20 g maltose, 165 mg $NH_4NO_3$, 750 mg glutamine, 100 mg myo-inositol, 0.4 mg thiamine HCl, 1.25 mg $CuSO_4 \cdot 5H_2O$, 3.5 g Phytagel, 50 mg hygromycin and 160 mg timentin. Each Petri dish (diameter 90 mm) contains 25 mL of culture medium.

### 2.4. RNA Extraction and Sequencing

RNA extraction was conducted using Plant RNA Extraction Kit (Takara, Japan), followed by assessment of RNA integrity using RNA Nano 6000 Assay Kit (Agilent Technologies, CA, USA). Then, cDNA library construction and RNA-Seq were conducted by LC-Bio Technologies (Hangzhou, China). Raw reads were trimmed to remove adapters, poly-N and low-quality reads. Clean reads were mapped to the reference genome of barley IBSC_v2 for transcriptome profiling. DESeq2 was used to calculate the expression levels of genes [34]. The genes with $|\mathrm{Log_2Fold\ Change}| \geq 1$ and false discovery rate (FDR) adjusted *p*-value < 0.01 were assigned as differentially expressed genes (DEGs).

### 2.5. Gene Validation and Expression Analysis

In this study, twelve DEGs were randomly selected for quantitative real-time PCR (qRT-PCR) to verify the data of RNA-Seq. qRT-PCR was conducted using TB Green® *Premix Ex Taq*™ II (Takara, Japan). The relative expression level was calculated using the $2^{-\Delta\Delta Ct}$ method with *Actin* as a housekeeping gene. All primers used for qRT-PCR were designed by the NCBI primer tools (http://www.ncbi.nlm.nih.gov/tools/primer-blast/ (accessed on 15 February 2022)), and listed in Table S1.

### 2.6. Statistical Analysis

The Venn diagrams, gene ontology (GO) and kyoto encyclopedia of genes and genomes (KEGG) analysis were performed using the OmicStudio tools (https://www.omicstudio.cn/tool (accessed on 15 February 2022)). Genes were annotated using several databases including Nr (NCBI non-redundant protein sequences), Nt (NCBI non-redundant nucleotide sequences), KOD (KEGG ortholog database) and GO (gene ontology).

## 3. Results

### 3.1. In Vitro Culture Performance of Immature Embryos

There was no significant difference between the two barley cultivars in the growth and morphology of immature embryos at early developmental stages. However, the distinct difference could be found in embryogenesis and explant regeneration at 11–12 weeks of in vitro culture. At the 80th day of in vitro culture, the immature embryos of GP had developed into hard, compact and nodular callus, indicating that embryogenic callus had been differentiated (Figure 1a), and these calli could continue in embryogenesis differentiation

and regenerate the complete explants. On the other hand, the immature embryos of ZU9 formed watery, soft and sticky callus, and no green plantlets could be observed (Figure 1b). Hence, these calli were considered as non-embryogenic ones, no ability for further differentiation and dying gradually.

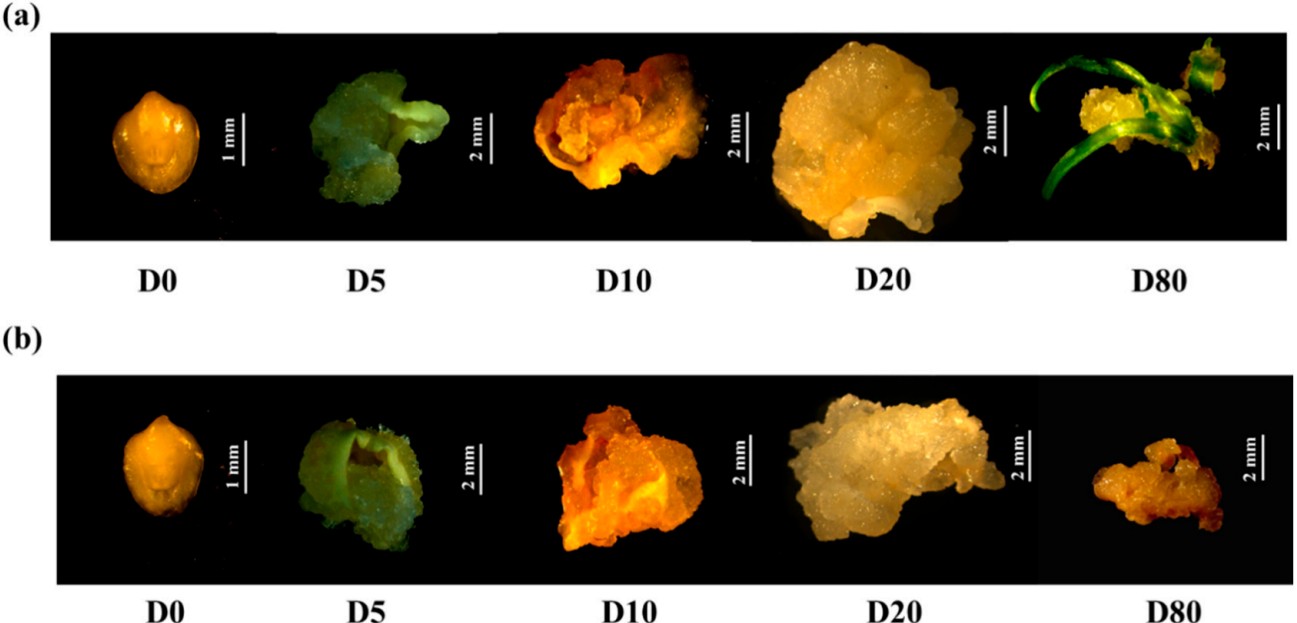

**Figure 1.** Callus morphology of the two barley genotypes at 0, 5, 10, 20, 80 days after tissue culture. (**a**) GP. (**b**) ZU9. D0, day 0; D5, day 5; D10, day 10; D20, day 20; D80, day 80.

*3.2. Global Analysis of Transcriptome Profiles in Barley Callus*

The constructed cDNA libraries were sequenced using Illumina paired-end sequencing technology. In total, 535,702,512 and 587,977,388 raw reads were obtained from the two cDNA libraries constructed from calli of GP and ZU9, respectively. The data quality check showed that more than 97% reads with Q30 > 97.98% were valid (Table S2), indicating sequencing results were of high quality and could be used for further analysis.

The differentially expressed genes (DEGs) in GP and ZU9 calli were identified by comparing the samples taken in the different culturing times with those taken at D0 as control ($|\text{Log}_2\text{Fold Change}| \geq 1$ FDR < 0.01). The number of DEGs in both GP and ZU9 calli increased with the culturing time. For GP, 2635, 3646 and 4796 genes were up-regulated and 2432, 2890 and 4541 down-regulated in D5, D10 and D20, respectively (Figure 2a). Moreover, 2972 genes were overlapped among three sampling dates (Figure 2b). For ZU9, 1901, 3126 and 4475 genes were up-regulated, and 3178, 4159 and 4798 down-regulated in D5, D10 and D20, respectively (Figure 2a). Similarly, 3015 genes were overlapped among three sampling dates (Figure 2c). Among these overlapped genes in GP and ZU9 calli, a total of 1172 (404 up-regulated and 768 down-regulated) and 1215 (823 up-regulated and 392 down-regulated) genes were specifically identified in GP and ZU9, respectively (Figures 2d and S2). The number of the overlapped genes was 1800 (Figure 2d), including 966 up-regulated and 834 down-regulated (Figure S2). GO enrichment showed that top 10 categories were same in these three gene groups (Figure S1). KEGG analysis revealed that the genes specifically identified in GP were mainly involved in "Plant hormone signal transduction", "MAPK signaling pathway-plant" and "Protein processing in endoplasmic reticulum", while the top 3 categories of genes specifically identified in ZU9 were "Plant-pathogen interaction", "Plant hormone signal transduction" and "Starch and sucrose metabolism" (Figure 2e,f). The overlapped genes were mainly enriched in "Plant hormone signal transduction", "Phenylpropanoid biosynthesis" and "Plant-pathogen interaction" (Figure 2g). More information on ZU9-specific DEGs and their overlapped genes were listed in Tables S4 and S5.

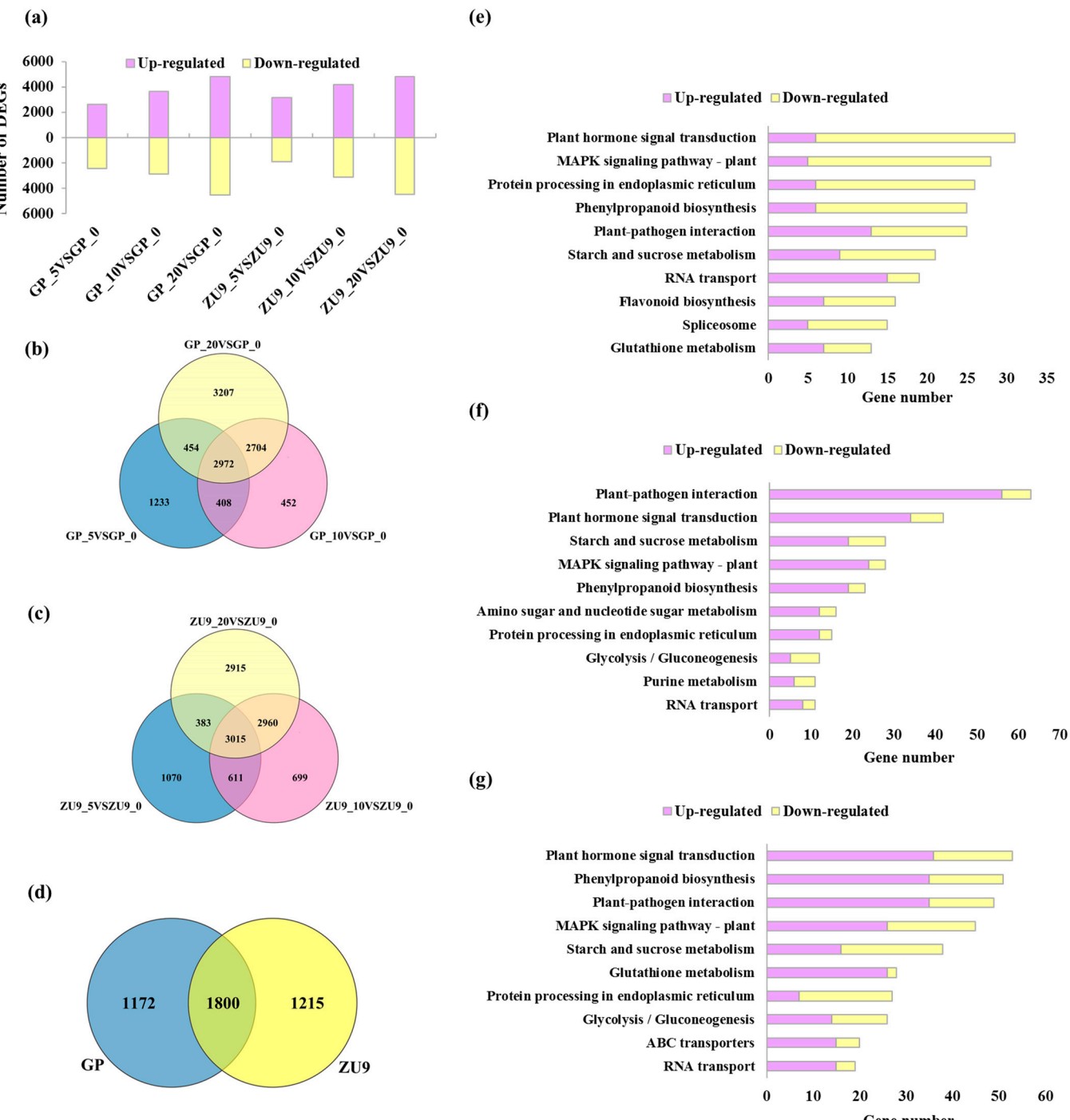

**Figure 2.** Summary of DEGs identified from the different samples. (**a**) The number of DEGs in the samples taken at different time. (**b**) DEGs in the GP calli at different times. (**c**) DEGs in ZU9 calli at different times. (**d**) Overlapped genes in GP and ZU9 calli. (**e**) The top 10 KEGG pathway categories in GP-specific DEGs. (**f**) The top 10 KEGG pathway categories in ZU9-specific DEGs. (**g**) The top 10 KEGG pathway categories in the common DEGs.

### 3.3. DEGs Involved in Biological Metabolism and Transcription Factors

We analyzed 1172 GP-specific DEGs and selected several genes with certain expression patterns for further examination. More specifically, these genes were significantly up-regulated or down-regulated in GP callus in all sampling dates, while they showed the contrast expression in ZU9 callus. On the whole, these genes were dramatically involved in transcription and translation, energy metabolism, stress and detoxification, protein kinase

and ubiquitin–proteasome system. For transcription, translation and energy metabolism, four genes encoding NAD(P)H dehydrogenase (*HORVU3Hr1G018530*), NAD(P)H oxidoreductase (*HORVU2Hr1G040130*), importin (*HORVU1Hr1G060180*), and eukaryotic translation initiation factor 3 (*HORVU5Hr1G098600*) were significantly up-regulated in GP, and showed much lower up-regulation level in ZU9 (Figure 3, Table S3).

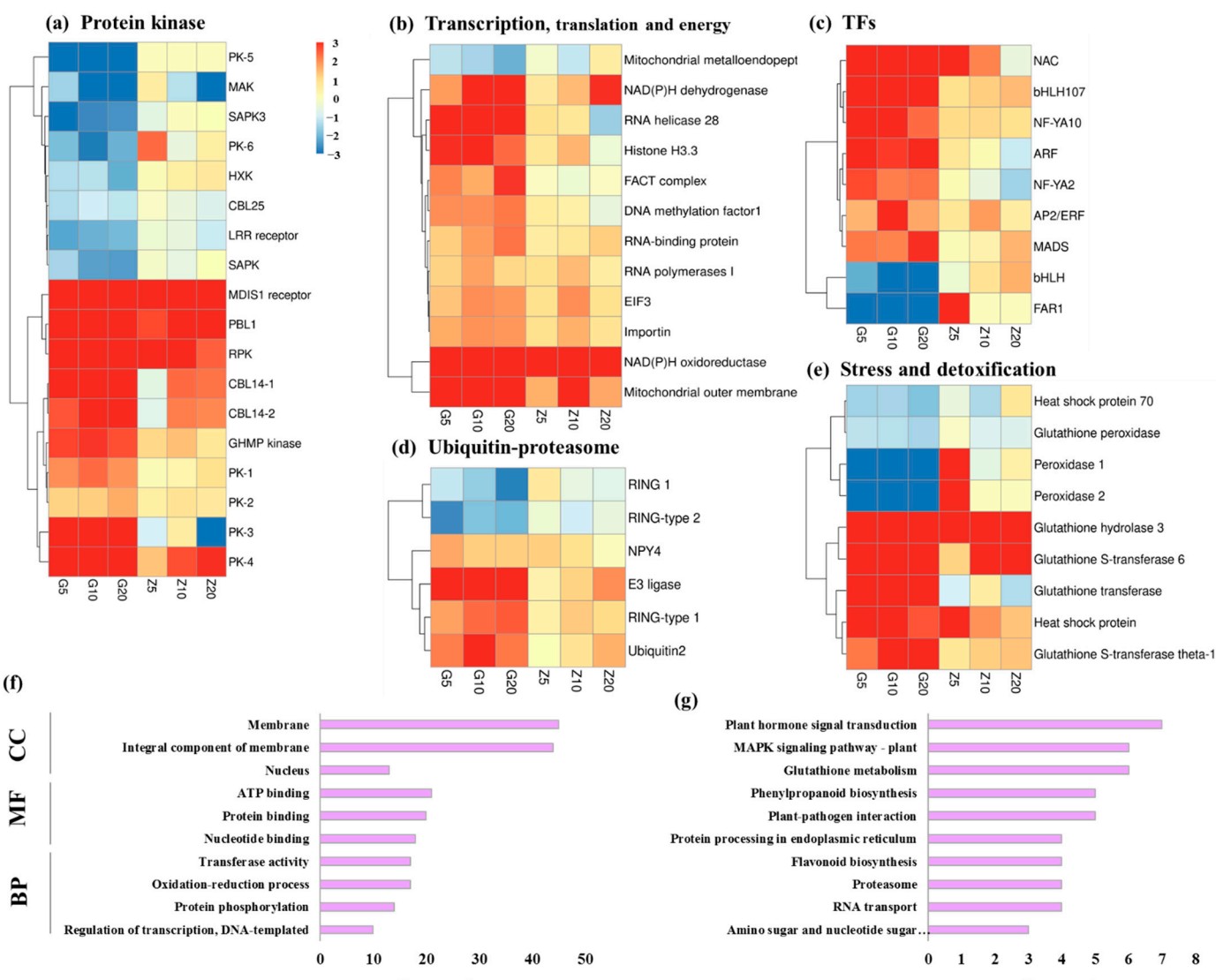

**Figure 3.** Heatmaps showing expression changes of some potential genes in GP-specific DEGs. (**a**) Protein kinases. (**b**) Transcription, translation and energy metabolism. (**c**) Transcription factors. (**d**) Ubiquitin–proteasome. (**e**) Stress and detoxification. (**f**) Top 10 categories of GO enrichment of these genes. 'BP', 'MF' and 'CC' represent 'biological process', 'molecular function' and 'cell component', respectively. (**g**) Top 10 categories of KEGG pathway of these genes.

For ubiquitin–proteasome, there were four genes encoding BTB/POZ domain-containing protein NPY4 (*HORVU4Hr1G013420*), E3 ubiquitin–protein ligase RING1 (*HORVU6Hr1G080320*), RING-type 1 (*HORVU1Hr1G051370*), and ubiquitin2 (*HORVU7Hr1G103060*), respectively. Similarly, their expression level was higher in GP than that in ZU9 at all sampling dates (Figure 3d, Table S3). Oppositely, *HORVU2Hr1G005630* and *HORVU5Hr1G071280* encoding RING-CH-type domain-containing protein and RING-type domain-containing protein were significantly down-regulated at D5 (Log$_2$Fold Change = −1.15, −2.70), D10 (−1.70, −1.89) and D20 (−2.81, −2.05) in GP, while they remained less changed at D5 (0.68, −0.35), D10 (−0.64, −1.04) and D20 (−0.73, −0.54) in ZU9. (Figure 3d, Table S3).

For stress and detoxification, there were four genes (*HORVU3Hr1G013650*, *HORVU3Hr1G112610*, *HORVU5Hr1G124630* and *HORVU5Hr1G118800*) involved in glutathione metabolism. All of them were significantly up-regulated at D5, D10 and D20 in GP, and the up-regulation was much lower in ZU9 (Figure 3e, Table S3). Oppositely, *HORVU6Hr1G063830* encoding glutathione peroxidase was down-regulated in GP, whereas its expression level was not significantly changed at D5 (−0.13), D10 (−0.86) and D20 (−0.76) in ZU9 (Figure 3e, Table S3). Meanwhile, *HORVU0Hr1G019500* and *HORVU0Hr1G015960* encoding peroxidase were significantly down-regulated at D5 (−3.00, −2.95), D10 (−4.03, −4.93) and D20 (−16.19, −13.28) in GP, while were significantly up-regulated by 3.20- and 8.92-fold at D5, and not significant at D10 and D20 in ZU9 (Figure 3e, Table S3).

There were 18 genes encoding different protein kinases and 9 genes encoding transcription factors (Figure 3a,c). For transcription factors, *NAC* (*HORVU6Hr1G075830*), *bHLH107* (*HORVU3Hr1G018680*), *AP2/ERF* (*HORVU3Hr1G030310*) and *MADS* (*HORVU6Hr1G002330*) were significantly up-regulated in GP, and showed much lower up-regulation level in ZU9 (Figure 3c, Table S3). Meanwhile, *bHLH* (*HORVU4Hr1G087590*) and *FAR1* (*HORVU3Hr1G080490*) were down-regulated at D5 (−2.18, −5.79), D10 (−4.06, −11.56) and D20 (−6.15, −11.56) in GP, whereas their expression change was not significant at all sampling dates except up-regulation of *HORVU4Hr1G087590* at D20 (1.58) in ZU9 (Figure 3c, Table S3). For protein kinases, *PBL1* (*HORVU1Hr1G037150*) and *RPK* (*receptor-like protein kinase*, *HORVU5Hr1G001000*) were significantly up-regulated at all sampling dates in GP, as well as *PK-1* (*protein kinase 1*, *HORVU3Hr1G089490*) and *PK-4* (*protein kinase 4*, *HORVU5Hr1G119270*), while their up-regulation level was much lower in ZU9 (Figure 3a, Table S3). Oppositely, *PK-5* (*protein kinase 5*, *HORVU1Hr1G025280*) and *SAPK3* (*serine/threonine-protein kinase*, *HORVU1Hr1G055340*) were significantly down-regulated at D5 (−12.23, −4.56), D10 (−6.37, −2.67) and D20 (−12.23, −2.53) in GP, and remained the less change in their expression in ZU9 (Figure 3a, Table S3).

For all these genes, GO term "molecular function" was largely enriched in "ATP binding", "protein binding" and "nucleotide binding" and "transferase activity". The top three categories of "cell component" were "membrane", "integral component of membrane" and "nucleus". For "biological progress", these genes were intensively involved in "oxidation-reduction process", "protein phosphorylation", and "regulation of transcription, DNA-templated" (Figure 3f). The KEGG pathway analysis showed that these DEGs were considerably gathered in "Plant hormone transduction", "MAPK signaling pathway-plant" and "Glutathione metabolism" (Figure 3g). Genes mentioned above showed different expression patterns between the two barley cultivars, suggesting that they might be associated with callus induction and formation.

### 3.4. Changes in the Expression of Phytohormone-Related Genes

Endogenous hormones play important roles in embryogenic callus induction and explant formation during tissue culture [20]. Therefore, we investigated the transcriptional profiling of the genes related to response, biosynthesis and transport of phytohormones, including auxin, cytokinin, gibberellin (GA), abscisic acid (ABA) and ethylene. For auxin, four *YUCCA*, one *PIN* (*PIN-FORMED*), one *TAR* (*tryptophan aminotransferase related*), two *SAUR* (*small auxin up RNA*) and three *IAA* (*indole-3-acetic acid*) were observed to be differentially expressed between the two groups (Table 1). More specifically, *YUCCA10* (*HORVU7Hr1G023880*) was up-regulated at D5 (7.90) and D20 (7.64) in GP. However, it was significantly down-regulated at all sampling dates (−6.34, −4.39, −11.28) in ZU9 (Table 1). Similarly, *SAUR36* (*HORVU5Hr1G085540*) was significantly down-regulated at all stages (−6.46, −6.46, −6.46) in ZU9, while it was up-regulated by 9.53-fold at D5 in GP (Table 1). *IAA13* (*HORVU5Hr1G094240*) was significantly up-regulated at all sampling dates (5.82, 2.28, 4.26) in GP while the change was not significant at D10 (0.10) and D20 (−0.47) in ZU9 (Table 1).

**Table 1.** Expression levels of phytohormone-related genes in callus of GP and ZU.

| Gene id | Description | GP | | | | ZU9 | | | |
|---|---|---|---|---|---|---|---|---|---|
| | | D5 [a] | D10 | D20 | Reg. [b] | D5 | D10 | D20 | Reg. |
| Auxin | | | | | | | | | |
| HORVU2Hr1G116980 | indole-3-pyruvate monooxygenase YUCCA5 | —— | —— | —— | Ns | −0.07 | −0.22 | −5.78 | Down |
| HORVU3Hr1G030390 | indole-3-pyruvate monooxygenase YUCCA6-like | −2.00 | −7.86 | −7.86 | Down | —— | —— | —— | Ns |
| HORVU5Hr1G028100 | indole-3-pyruvate monooxygenase YUCCA6-like | −8.25 | −8.25 | −1.45 | Down | —— | —— | —— | Ns |
| HORVU7Hr1G023880 | indole-3-pyruvate monooxygenase YUCCA10 | 7.90 | —— | 7.64 | Up | −6.34 | −4.39 | −11.28 | Down |
| HORVU1Hr1G072230 | tryptophan aminotransferase related 2 | 0.72 | −0.64 | −1.53 | Ns | 1.87 | 0.53 | 1.03 | Up |
| HORVU3Hr1G078620 | putative auxin efflux carrier component 5 | —— | —— | —— | Down | 6.92 | 6.07 | —— | Up |
| HORVU2Hr1G119400 | auxin-responsive protein IAA14-like | −0.18 | −0.47 | −3.61 | Down | 2.05 | 2.87 | 1.19 | Up |
| HORVU5Hr1G081180 | auxin-responsive protein IAA26-like | —— | —— | 7.29 | Up | −8.43 | −8.43 | −8.43 | Down |
| HORVU5Hr1G094240 | auxin-responsive protein IAA13-like | 5.82 | 2.28 | 4.26 | Up | 3.78 | 0.10 | −0.47 | Down |
| HORVU5Hr1G076740 | auxin-responsive protein SAUR50-like | 7.45 | —— | —— | Up | 2.09 | 0.66 | −1.11 | Down |
| HORVU5Hr1G085540 | auxin-responsive protein SAUR36 | 9.53 | —— | —— | Up | −6.46 | −6.46 | −6.46 | Down |
| Cytokinin | | | | | | | | | |
| HORVU3Hr1G063960 | adenylate isopentenyl transferase 1 | 11.67 | 9.67 | 10.55 | Up | 3.06 | −0.48 | −0.82 | Down |
| HORVU0Hr1G005240 | cytokinin dehydrogenase 8-like | 11.54 | —— | —— | Ns | —— | —— | 11.38 | Up |
| HORVU1Hr1G057860 | cytokinin dehydrogenase 2 | —— | 4.98 | 4.71 | Up | −0.91 | −1.50 | −2.27 | Down |
| HORVU2Hr1G090140 | cytokinin dehydrogenase 8-like | −8.29 | −8.29 | −3.40 | Down | —— | 5.48 | —— | Up |
| HORVU3Hr1G027460 | cytokinin oxidase/dehydrogenase | −8.58 | −8.58 | −8.58 | Down | —— | —— | —— | Ns |
| HORVU6Hr1G039680 | cytokinin oxidase/dehydrogenase | —— | —— | —— | Ns | −7.12 | −7.12 | −7.12 | Down |
| HORVU6Hr1G039690 | cytokinin oxidase/dehydrogenase | —— | —— | —— | Ns | −8.38 | −8.38 | −8.38 | Down |
| HORVU7Hr1G086710 | cytokinin dehydrogenase 10-like | —— | —— | 7.77 | Up | −6.64 | −6.64 | −6.64 | Down |
| HORVU2Hr1G064940 | histidine kinase 5 | −0.71 | −0.48 | −9.35 | Down | —— | —— | —— | Ns |
| HORVU3Hr1G069130 | histidine kinase 1 | 7.31 | 7.54 | 6.54 | Up | —— | —— | —— | Ns |
| HORVU5Hr1G012530 | histidine kinase 4 | −2.91 | 2.59 | 4.53 | Up | −2.10 | −1.86 | −1.50 | Down |
| HORVU4Hr1G001670 | histidine-containing phosphotransfer protein 2-like | —— | 10.32 | 11.00 | Up | —— | —— | —— | Ns |
| HORVU4Hr1G074350 | histidine-containing phosphotransfer protein 2-like | —— | 8.67 | 11.46 | Up | —— | —— | —— | Ns |
| HORVU7Hr1G049600 | histidine-containing phosphotransfer protein 2-like | 7.30 | 9.62 | —— | Up | —— | —— | —— | Ns |
| HORVU7Hr1G114450 | two-component response regulator ARR2 | —— | 6.79 | 6.40 | Up | —— | —— | —— | Ns |
| Gibberellin | | | | | | | | | |
| HORVU5Hr1G082380 | ent-kaur-16-ene synthase, chloroplastic isoform X1 | —— | —— | —— | Ns | 8.59 | 10.13 | 6.24 | Up |
| HORVU7Hr1G101720 | gibberellin 2-beta-dioxygenase 6-like | —— | —— | 7.25 | Up | −9.98 | −9.98 | −2.20 | Down |
| HORVU1Hr1G090800 | DELLA protein SLR1-like | —— | —— | 5.28 | Up | −6.81 | −6.81 | −0.50 | Down |

**Table 1.** *Cont.*

| Gene id | Description | GP | | | | ZU9 | | | |
|---|---|---|---|---|---|---|---|---|---|
| | | D5 [a] | D10 | D20 | Reg. [b] | D5 | D10 | D20 | Reg. |
| Abscisic acid | | | | | | | | | |
| HORVU1Hr1G074200 | putative aldehyde oxidase-like protein | 10.37 | 9.08 | —— | Up | 1.30 | −1.71 | −7.62 | Down |
| HORVU3Hr1G062290 | aldehyde oxidase GLOX-like | —— | —— | —— | Ns | −6.48 | −6.48 | −6.48 | Down |
| HORVU4Hr1G019740 | aldehyde oxidase GLOX-like | 1.09 | 0.66 | −2.16 | Down | 11.44 | 9.54 | 8.83 | Up |
| HORVU0Hr1G032390 | abscisic acid receptor 8 | 11.57 | —— | 10.34 | Up | −0.64 | 0.23 | −10.94 | Down |
| HORVU3Hr1G040680 | abscisic acid receptor PYL2-like | 10.61 | —— | 8.75 | Up | 0.37 | −1.08 | −8.88 | Down |
| HORVU7Hr1G029040 | putative protein phosphatase 2C 6 | —— | 7.12 | 7.06 | Up | −0.75 | −0.29 | −1.30 | Down |
| HORVU0Hr1G032840 | serine/threonine-protein kinase SAPK4-like | —— | —— | 7.32 | Up | —— | —— | —— | Ns |
| Ethylene | | | | | | | | | |
| HORVU3Hr1G098750 | ACC synthase | —— | —— | —— | Ns | −3.93 | −4.65 | −0.97 | Down |
| HORVU0Hr1G001750 | ethylene receptor, partial | 7.19 | —— | —— | Up | −9.24 | −1.07 | −9.24 | Down |
| HORVU3Hr1G093140 | serine/threonine-protein kinase CTR1 | 0.79 | 0.42 | −3.29 | Ns | 2.67 | 2.12 | 2.07 | Up |
| HORVU2Hr1G098250 | ethylene-responsive transcription factor ERF105-like | 0.91 | 1.45 | 0.21 | Ns | 0.68 | 2.63 | 2.72 | Up |
| HORVU4Hr1G015350 | ethylene-responsive transcription factor ERF022-like | 1.74 | 2.19 | 0.55 | Up | 6.02 | 4.67 | 4.23 | Up |

[a] The value in the table represents $Log_2$(Fold Change) of genes at different stages (D5, D10 and D20) compared with D0. [b] Reg.: regulation patterns. Up: upregulation, Down: downregulation, Ns: not significant.

For cytokinin, one *IPT* (adenylate isopentenyl transferase), seven *CKX* (cytokinin oxidase/dehydrogenase), three *AHK* (*Arabidopsis* histidine kinase), three *AHP* (*Arabidopsis* histidine phosphotransfer protein), and one ARR (*Arabidopsis* response regulator) were found between the two groups (Table 1). In detail, *ARR2* (*HORVU7Hr1G114450*) and *AHP2* (*HORVU4Hr1G001670*) were significantly up-regulated at D10 (10.32, 6.79) and D20 (11.00, 6.40) in GP while both of their expression level was not significantly changed at all sampling dates in ZU9 (Table 1). Besides, *IPT1* (*HORVU3Hr1G063960*) was significantly up-regulated at D5 (11.67), D10 (9.67) and D20 (10.55) in GP. However, it showed no significant change at D10 ($-0.48$) and D20 ($-0.82$) in ZU9 (Table 1).

The number of the genes associated with GA, ABA and ethylene was much less than those of auxin and cytokinin. In detail, only *KO* (*ent-kaurene oxidase*, *HORVU5Hr1G082380*), *KAO* (*ent-kaurene acid oxidase*, *HORVU7Hr1G101720*) and *DELLA* (*HORVU1Hr1G090800*) were detected in terms of GA (Table 1). Both of *KAO* (*HORVU7Hr1G101720*) and *DELLA* (*HORVU1Hr1G090800*) were only up-regulated at D20 (7.25, 5.28) in GP whereas they were both down-regulated at all sampling dates in ZU9 except *DELLA* (*HORVU1Hr1G090800*) at D20 ($-0.50$) (Table 1). For ABA, there were three *AAO* (*aldehyde oxidase*), two *PYL* (*pyrabactin resistance1-like*), one *PP2C* (*protein phosphatase 2C*) and one *SAPK* (*serine/threonine-protein kinase*) (Table 1). *PYL8* (*HORVU0Hr1G032390*) and *PYL2* (*HORVU3Hr1G040680*) were up-regulated at D5 (11.57, 10.61) and D20 (10.34, 8.75) while they were down-regulated by 10.94- and 8.88-fold at D20 in ZU9, respectively (Table 1). Finally, for ethylene, one *ACS* (*1-Aminocyclopropane-1-carboxylic acid synthase*), one *ETR* (*ethylene receptor*), one *CTR1* (*constitutive triple response1*) and two *ERF* (*ethylene-responsive factor*) were detected in the two groups (Table 1). *ERF105* (*HORVU2Hr1G098250*) and *ERF022* (*HORVU4Hr1G015350*) were significantly up-regulated at all sampling dates in ZU9 except *ERF105* (*HORVU2Hr1G098250*) at D5 (0.68), whereas they remained less change in GP (Table 1).

In addition, twelve DEGs were randomly selected for qRT-PCR to confirm the results of transcriptome analysis. The results from qRT-PCR were consistent with those of RNA-Seq (Figure 4a) and significantly positive correlation was observed between them ($r^2 = 0.814$) (Figure 4b), indicating the accuracy of RNA-Seq data.

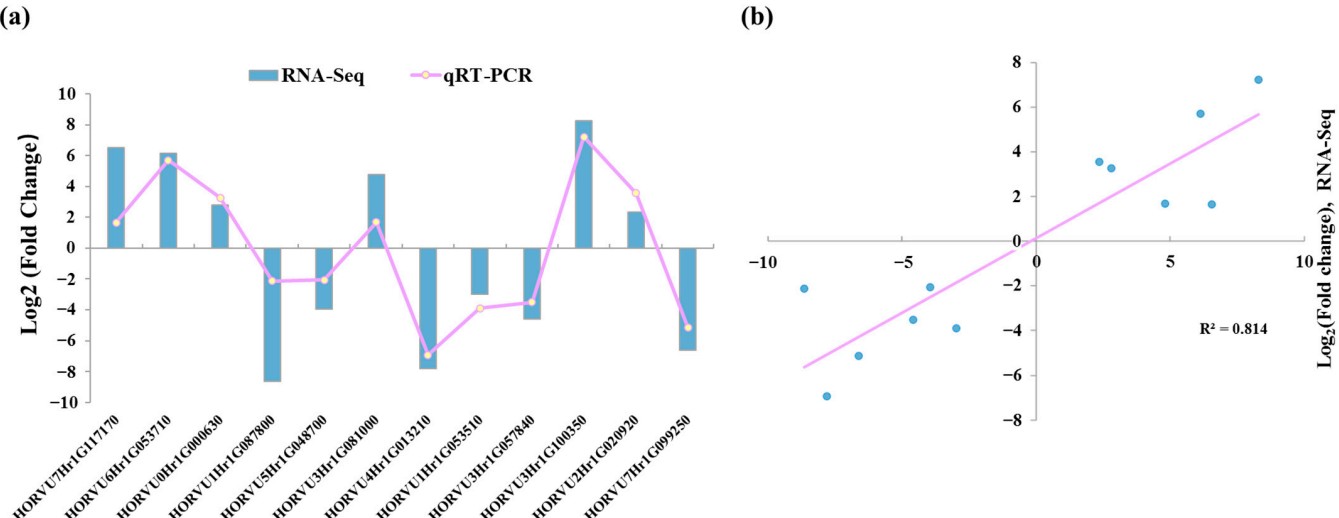

**Figure 4.** qRT-PCR analysis and the correlation between the results of RNA-Seq and qRT-PCR. (**a**) The relative expression level was calculated using the $2^{-\Delta\Delta Ct}$ method with Actin as a housekeeping gene. (**b**) The correlation between the Log$_2$(Fold Change) of twelve DEGs of RNA-Seq (*x* axis) and qRT-PCR (*y* axis).

## 4. Discussion

Barley is more recalcitrant to tissue culture than other cereal crops [35]. In this study, we compared global transcriptional changes of cultured immature embryos of two barley

genotypes taken from different stages of callus induction, and identified several potential factors associated with the genotypic difference in embryogenesis differentiation and plantlet regeneration.

The genetic transformation in plants is largely dependent on the ability of callus induction and plantlet regeneration, which show a great variation among species and genotypes [36]. In barley, GP is widely used in genetic transformation due to its high tissue culture capacity [37,38]. However, GP was an old cultivar with unsatisfactory agronomic traits in Europe, and only some distilleries still use it for whisky production now [31]. More barley cultivars are expected to be used in genetic transformation, but few cultivars are available because of low callus induction and plantlet regeneration. Moreover, the reason why there is a distinct difference in the capacity of callus induction and plantlet regeneration among genotypes is still elusive. In this experiment, two cultivars, GP and ZU9 were compared in their capacity of callus induction, differentiation and plantlet formation, and the results showed that immature embryos of ZU9 and GP had the different destiny in callus induction and plantlet regeneration (Figure 1). As expected, the calli of GP could continue to form green plantlet, while the calli of ZU9 stopped further differentiation. Hisano et al. [20] used immature embryos of three malting barley cultivars, GP, Haruna Nijo and Morex, to evaluate regeneration ratios and they found immature embryos of GP had the highest ratio of green plantlet regeneration, being consistent with our results. Han et al. [39] compared the plantlet regeneration of 12 barley genotypes including GP, and found Zaoshu 3 had relatively high regeneration ability, although it was not as high as GP. Currently the genetic transformation efficiency of GP is considered the highest among the examined barley genotypes [35,40].

The genes related to protein metabolism are very active in embryogenic callus cells with dense cytoplasm and prominent nuclei, indicating that the activity of transcription and translation is higher in these callus cells than that in non-embryogenic callus cells [41]. In the present study, the genes involved in DNA and RNA metabolism, such as RNA binding protein (*HORVU2Hr1G092920*) and RNA helicase 28 (*HORVU7Hr1G111810*), were significantly up-regulated in GP calli at various stages (Figure 3b). Besides, importin (*HORVU1Hr1G060180*) facilitating RNA transport into nucleus by binding specific recognition factors was abundant, and up-regulation of *eIF3* (*HORVU5Hr1G098600*) was also observed in GP calli (1.35, 2.02, 1.86), demonstrating that protein metabolism represents a key event in embryogenesis, particularly at early stage [42]. In other species, ribosomal proteins were identified during embryo development [43], suggesting the importance of high translation activity. In addition, we found some up-regulated genes encoding NAD(P)H dehydrogenase (*HORVU3Hr1G018530*), NAD(P)H oxidoreductase (*HORVU2Hr1G040130*), mitochondrial outer membrane protein (*HORVU3Hr1G067610*) in GP (Figure 3b), which were involved in mitochondrial metabolism. Their up-regulation level was not much lower in ZU9 (Figure 3b, Table S3), suggesting that they played a crucial part in embryogenesis by providing energy for basic metabolism in embryogenic callus cells [44].

Ubiquitin proteins are required for recycle and degradation of misfolded and damaged proteins [45]. In this study, *Ubiquitin 2* (*HORVU7Hr1G103060*) was evidently up-regulated in GP at all sampling dates (2.16, 3.61, 2.22) (Figure 3d, Table S3), making the denatured proteins to proteasomes. In wheat, TaMAB2 interacts with eIF3/4 by E3 ligase, regulating translation initiation during the onset of embryogenesis [46]. Significant up-regulation of the genes encoding BTB/POZ domain-containing protein (*HORVU4Hr1G013420*) and E3 ubiquitin–protein ligase RING1 (*HORVU6Hr1G080320*) in GP and less fold change in ZU9 demonstrated a high translation rate in embryogenic calli (Figure 3d).

In general, the genes involved in stress and detoxification are often linked with embryogenesis, as their high expression facilitates the programmed cell death and cell de-differentiation [47,48]. Up-regulation of *HSP* (*heat shock protein*, *HORVU2Hr1G046370*) in GP calli (5.04, 4.40, 2.59) suggested the occurrence of correct folding for the synthesized proteins and refolding of the denatured or unstable proteins in response to in vitro tissue culture condition (Figure 3e, Table S3), which acts a stress for the callus cells [49]. Other members

of heat-shock proteins, such as HSP60 and HSP70, were also reported to participate in development of embryogenic calli in other species [50–52]. It can be indicated that these sorts of genes are favorable for correcting protein functions and reducing the occurrence of misfolded proteins under various stresses at early stage of callus induction [53].

Exogenous application of plant growth regulators in tissue culture system often result in accumulation of cellular toxins, which need to be detoxified in cells [43]. Glutathione-S-transferases (GSTs) catalyze the conjugation of the tripeptide glutathione (GSH) and act as GSH-dependent peroxidases [54,55]. Cells with higher activity of these enzymes are more tolerant to auxin, which is an essential hormone used in callus induction [56]. Therefore, the high abundance of GSTs in embryogenic callus cells is related to great detoxification. In this study, higher expression level of four genes related to glutathione S-transferase (*HORVU3Hr1G013650*, *HORVU3Hr1G112610*, *HORVU5Hr1G124630* and *HORVU5Hr1G118800*) was observed in GP calli than that in ZU9 at the various culture stages (Figure 4e, Table S3), indicating the importance of antioxidant enzymes in embryogenesis.

Auxin has been long identified as an efficient inducer for callus induction and formation [57]. YUCCA is a rate-limiting enzyme in auxin biosynthesis pathway and catalyzes the last step of indole-3-acetic acid (IAA) from indole-3-pyruvate [58]. *IAA* encodes auxin-responsive protein and *SAUR* functions as a positive factor in cell expansion through the modulation of auxin transport [59]. In this study, the results showed that auxin signaling pathway differed in the two genotypes. *YUCCA10* (*HORVU7Hr1G023880*), the rate-limiting enzyme in the biosynthesis of auxin, was up-regulated at D5 (7.90) and D20 (7.64) in GP with significant down-regulation at all sampling dates (6.34, 4.39, 11.28) in ZU9 (Table 1), possibly leading to different auxin concentration in GP and ZU9 (Figure 5). *IAA13* (*HORVU5Hr1G094240*) was significantly up-regulated at all sampling dates (5.82, 2.28, 4.26) in GP, interacting with *SAUR36* (*HORVU5Hr1G085540*), which was also up-regulated by 9.53-fold in GP, whereas they exhibited down-regulation or were unchanged in ZU9 during callus induction (Table 1; Figure 5). It was observed that two auxin-related genes (*SAUR* and *SAUR50*) were up-regulated in the barley genotypes with high callus formation, and down-regulated in the recalcitrant genotype [60]. Downstream auxin response factors (ARFs) play important roles in callus formation by modifying cell wall or regulating cell cycle [23,24]. ARF7 and ARF19 could activate the expression of LBD family, mediating callus formation [21,61,62]. Previous studies also revealed that the auxin receptor TIR1 and efflux carrier PIN-FORMED (PIN) participated in shoot formation from calli [36,63]. These results indicated that the change in expression of the genes involved in biosynthesis, response and transport of auxin might lead to auxin content gradients, which is associated with cell reprogramming during callus induction.

Cytokinin is also considered as a crucial endogenous molecule determining cell fate and pattern during plant regeneration [64,65]. Several genes have been found to be associated with biosynthesis and signaling pathways of cytokinin. *Adenylate isopentenyl transferases* (*IPTs*), encoding the rate-limiting enzyme for cytokinin biosynthesis, play a crucial part in green plantlet formation [66]. Cytokinin receptors, *Arabidopsis* histidine kinases (AHKs), were identified to positively facilitate shoot formation by promoting the expression of *WUSCHEL* (*WUS*) [67]. Additionally, the transcription factor *Arabidopsis* response regulator 2 (ARR2) is required for the protective activity of cytokinin [68]. In this study, *IPT1* was significantly up-regulated in GP at different stages (11.67, 9.67, 10.55) while it showed no significant change at D10 (−0.48) and D20 (−0.82) in ZU9 (Table 1; Figure 5). Moreover, it was observed that *AHP2* (*HORVU4Hr1G001670*) and *ARR2* (*HORVU7Hr1G114450*) were significantly up-regulated at D10 (10.32, 6.79) and D20 (11.00, 6.40) in GP while both of their expression level was not significantly changed at all sampling dates in ZU9 (Table 1; Figure 5). Therefore, we suggest that the contrasting expression levels of *IPT1*, *AHP2* and *ARR2* in GP and ZU9 might cause different in vitro culture performance of their calli.

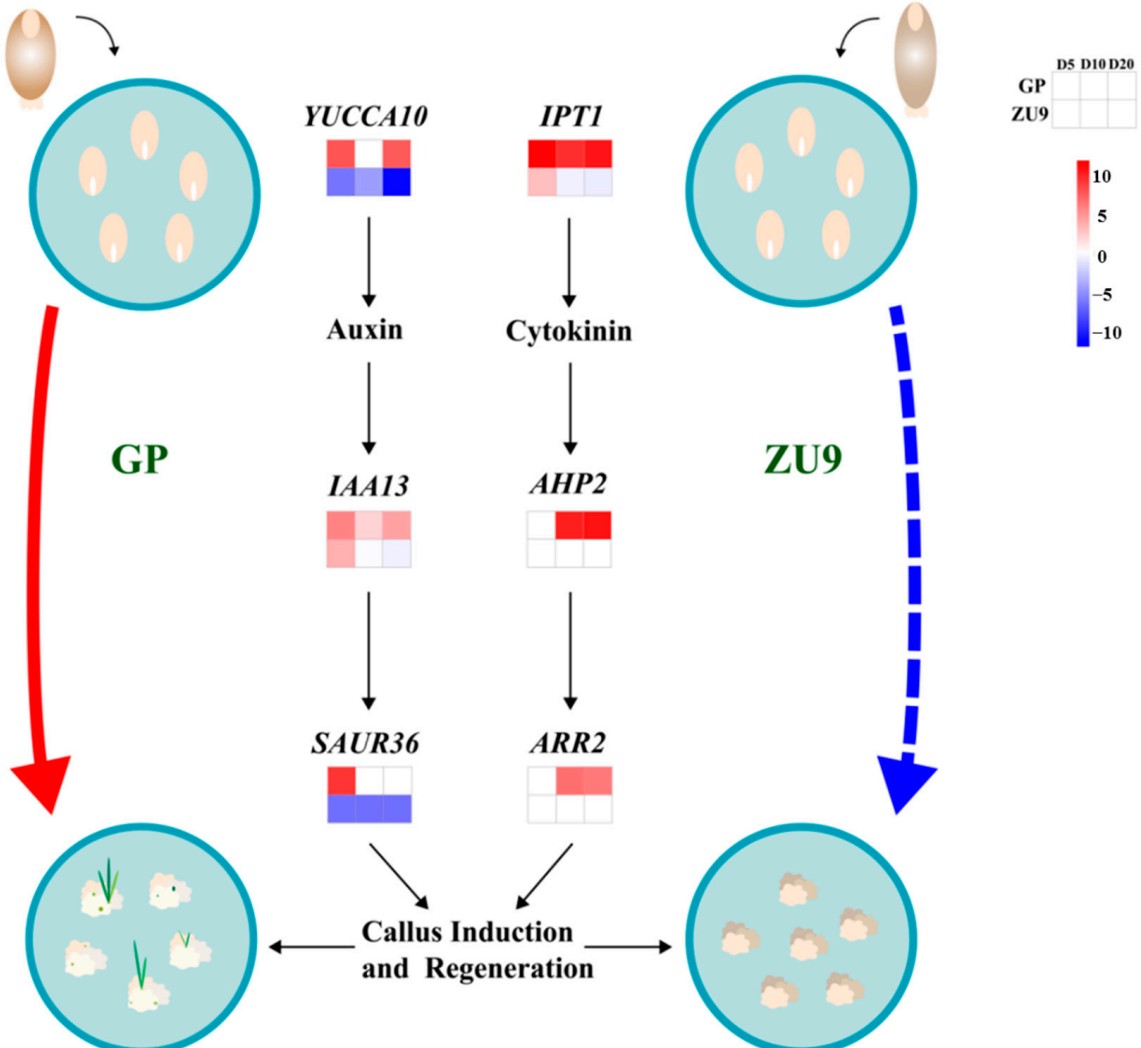

**Figure 5.** A proposed regulatory network of DEGs related to biosynthesis, response and transport of auxin and cytokinin in response to in vitro culture of two barley explants. IPT1: isopentenyltransferase1; AHP2: *Arabidopsis* histidine phosphotransfer protein 2; ARR2: *Arabidopsis* response regulator 2; IAA13: indole-3-acetic acid 13; SAUR36: small auxin up RNA 36. Red marks mean positive or up-regulated expression patterns; blue marks mean negative or down-regulated expression patterns.

## 5. Conclusions

Barley is the fourth worldwide cereal crop and is relatively recalcitrant to tissue culture than some other cereal crops. To reveal the key factors controlling callus induction in barley, we compared the transcriptome profiles of in vitro cultured immature embryos of two barley genotypes, Golden Promise (an old cultivar widely used in genetic transformation) and ZU9 (a malting cultivar being planted in Zhejiang, China), differing largely, in the efficiency of genetic transformation, so as to understand the reasons why Golden Promise is more amenable to stable transformation than ZU9. A total of 5386 up-regulated and 6257 down-regulated DEGs were identified in GP, respectively. From them, several genes identified exclusively in GP were selected and highlighted for further investigation. They were mainly involved in protein metabolism, energy metabolism, stress response, detoxification and ubiquitin–proteasome. Furthermore, we proposed a regulatory network of DEGs related to biosynthesis, response and transport of auxin and cytokinin in response to in vitro culture of the two barley calli. The comprehensive analysis provides insights into the molecular mechanisms of callus induction in barley and are helpful for improving genetic transformation of barley as well as other cereal crops.

**Supplementary Materials:** The following are available online at https://www.mdpi.com/article/10.3390/agronomy12030749/s1, Figure S1: Top 10 Gene Ontology (GO) categories of DEGs in different groups, Figure S2: The number of up- and down-regulated genes in three groups (GP-Specific, ZU9-Specific and Overlap), Table S1: Primers for qRT-PCR in this study, Table S2: Evaluation results of sample-sequencing data, Table S3: Selected genes with contrasting expression level in GP and ZU9, Table S4: DEGs specifically identified in ZU9, Table S5: Common DEGs between overlapped DEGs of GP and ZU9 at different sampling dates.

**Author Contributions:** Conceptualization and methodology, Q.S. and G.Z.; Investigation, Z.X., F.W., Y.T. and Y.X.; Formal analysis, Z.X., F.W., Y.T. and Y.X.; Writing—original draft preparation, Z.X.; Writing—review and editing, Q.S., G.Z. and Z.X. All authors have read and agreed to the published version of the manuscript.

**Funding:** This work was supported by Natural Science Foundation of China (32171929, 31901429), China Agriculture Research System (CARS-05), Key Research Foundation of Science and Technology Department of Zhejiang Province of China (2021C02064-3), Key Research Projects of Zhejiang Province (2021C02057) and Jiangsu Collaborative Innovation Center for Modern Crop Production (JCIC-MCP).

**Institutional Review Board Statement:** Not applicable.

**Informed Consent Statement:** Not applicable.

**Data Availability Statement:** The data presented in this study are available in the article and Supplementary Materials.

**Acknowledgments:** This work was supported by Natural Science Foundation of China (32171929, 31901429), China Agriculture Research System (CARS-05), Key Research Projects of Zhejiang Province (2021C02057) and Jiangsu Collaborative Innovation Center for Modern Crop Production (JCIC-MCP).

**Conflicts of Interest:** The authors declare no conflict of interest.

## Abbreviations

2,4-D: 2,4-Dichlorophenoxyacetic acid; 6-BA: N-(Phenylmethyl)-9H-purin-6-amine; AAO: aldehyde oxidase; ABA: abscisic acid; ACS: 1-Aminocyclopropane-1-carboxylic acid synthase; AHK: *Arabidopsis* histidine kinase; AHP: *Arabidopsis* histidine phosphotransfer protein; ARF: auxin response factor; ARR: *Arabidopsis* response regulator; CI: callus induction; CIS: callus induction selection; CKX: cytokinin oxidase/dehydrogenase; CTR1: constitutive triple response1; DEG: differentially expressed genes; eIF: eukaryotic initiation factor; ERF: ethylene-responsive factor; ETR: ethylene receptor; FDR: false discovery rate; FC: fold change; GA: gibberellin; GSH: glutathione; GSTs: glutathione-S-transferases; GO: gene ontology; GP: Golden Promise; HSP: heat shock protein; IAA: indole-3-acetic acid; IPT: adenylate isopentenyl transferase; KAO: ent-kaurene acid oxidase; KEGG: kyoto encyclopedia of genes and genomes; KO: ent-kaurene oxidase; KOD: KEGG ortholog database; LBD: lateral organ boundaries domain; Nr: NCBI non-redundant protein sequences; Nt: NCBI non-redundant nucleotide sequences; PIN: PIN-FORMED; PK-1: protein kinase-1; PK-4: protein kinase-4; PK-5: protein kinase-5; PP2C: protein phosphatase 2C; PYL: pyrabactin resistance1-like; qRT-PCR: quantitative real-time PCR; RNA-Seq: RNA sequencing; RPK: receptor-like protein kinase; SAPK: serine/threonine-protein kinase; SAUR: small auxin up RNA; TAR: tryptophan aminotransferase related; TFs: transcription factors; YUC: YUCCA flavin monooxygenases.

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
