# Peer review of "Transcriptome Analysis Reveals Genetic Factors Related to Callus Induction in Barley"

_agronomy, doi:10.3390/agronomy12030749_

Round 1

Reviewer 1 Report

In the current study, Zhengyuan Xu et al. made a very nice tail to address some major questions about the transformation efficiency of barley via transcriptome analysis. They have analyzed and interpreted a bit large sets of various data categories. Well presented and clear figures. Nicely written discussion covering all results, and answering all raised questions across the manuscript. 

However, there are some comments:

Major comments:

1- The title can be changed to avoid the word "control" it can be replaced by "related to" or something else, because they did not identify any genes that are intensively controlling the callus development or differentiation. 

2- The authors claimed that these genes are controlling the callus differentiation, however, they did not measure any morphological or physiological parameters that indicate the callus development status in both objective barley genotypes.

3- The introduction section can be more informative, if some more literatures are cited  such as 

https://www.sciencedirect.com/science/article/pii/S0734975019301843

https://link.springer.com/protocol/10.1007/978-1-4939-8944-7_8

More literatures illustrating the functions of the different categories of genes in callus development. You can do that by citing some references from the discussion section in the introduction section. 

 Minor comments:

a- The abstract section includes so many abbreviations making the read not joyfull for the reader

b- Line 46, numbers from one to twelve should be written in letters

c- Line 60: Always start the sentences with the full name of the plant or genotype, do not start with abbreviation, especially if this is not well known such as DNA

d- Line 71: calli not callus

e- Line 81: After sterilized " I think that this sentence might be be rephrased "

f- Line 101: Insert space after 2.7

g- Line 169: I think that the word were makes more sense than was in this sentence.

h- In figure 1a, b and c: the genotypes names/abbreviation (G_ and Z_) should be consistent with figures legends  and text (GP and ZU.

i- Line 194: less changed not less change

j- Line 232: the two barley not two barley 

k- In the first sentence of the discussion section the authors cited a reference (Lim et al. 2014, ref. no. 29) stating Barley is more recalcitrant to tissue culture than the remaining cereals, however, they cited another reference in the introduction section stating that wheat is more recalcitrant one (Hiei et al. ref. 29). The authors can combine both reference in this sentence like this "Barley and wheat are more recalcitrant to tissue culture (18, 29), this makes  more sense.  

Line 1, Page 10: the genes names should be italic across the whole manuscript some were not like this one  HORVU7Hr1G114450 and others, especially in first paragraph of page 10 " directly after table 1". Please, revise that across the manuscript

L- Line 232,  Page 13, : Arabidopsis sometimes italic/regular? please, be consistent

m- Line 166, Page 14: again "the two barley not two barley"

Author Response

To Reviewer 1

Major comments:

1- The title can be changed to avoid the word "control" it can be replaced by "related to" or something else, because they did not identify any genes that are intensively controlling the callus development or differentiation.

Answer 1: Thanks for your suggestion. It is true that we did not identify the function of candidate genes directly. Hence, the title has been changed, as you suggested (Line 1, Page 1).

2- The authors claimed that these genes are controlling the callus differentiation, however, they did not measure any morphological or physiological parameters that indicate the callus development status in both objective barley genotypes.

Answer 2: It is true that we did not measure any morphological or physiological parameters of the calli. However, the distinct difference could be directly observed in embryogenesis and explant regeneration at 11-12 weeks of in vitro culture between the two barley genotypes. The immature embryos of GP developed into hard, compact and nodular callus with green plantlets, indicating that embryogenic callus have differentiated, while those of ZU9 become watery, soft and sticky callus, without green plantlets being observed.

3- The introduction section can be more informative, if some more literatures are cited such as

https://www.sciencedirect.com/science/article/pii/S0734975019301843

https://link.springer.com/protocol/10.1007/978-1-4939-8944-7_8

More literatures illustrating the functions of the different categories of genes in callus development. You can do that by citing some references from the discussion section in the introduction section.

Answer 3: Thanks for your comment. We cited more relevant literatures in the introduction, including these two references mentioned above (Line 35 and 57-65, Page 1-2).

 Minor comments:

a- The abstract section includes so many abbreviations making the read not joyfull for the reader.

Answer a: Thanks for your suggestion and we deleted some abbreviations in the introduction (Line 17-21, Page 1).

b- Line 46, numbers from one to twelve should be written in letters

Answer b: Sorry for that, and we have corrected it (Line 49, Page 2).

c- Line 60: Always start the sentences with the full name of the plant or genotype, do not start with abbreviation, especially if this is not well known such as DNA

Answer c: Thanks for your comment, we made the change (Line 72, Page 2).

d- Line 71: calli not callus

Answer d: Thank you for your careful reading, and we have corrected it (Line 82, Page 2).

e- Line 81: After sterilized " I think that this sentence might be rephrased "

Answer e: Based on your comment, we rephased the sentence (Line 97-98, Page 3).

f- Line 101: Insert space after 2.7

Answer f: We have inserted space after 2.7 (Line 118, Page 3).

g- Line 169: I think that the word ‘were’ makes more sense than ‘was’ in this sentence.

Answer g: The word ‘was’ has been replaced by ‘were’, as you suggested (Line 187, Page 5).

h- In figure 1a, b and c: the genotypes names/abbreviation (G_ and Z_) should be consistent with figures legends and text (GP and ZU).

Answer h: Sorry for our negligence. We remade the figure (Figure 2).

i- Line 194: less changed not less change

Answer i: We have corrected it in the text (Line 215, Page 6).

j- Line 232: the two barley not two barley

Answer j: We added ‘the’ before ‘two barley’ in the text (Line 251, Page 7).

k- In the first sentence of the discussion section the authors cited a reference (Lim et al. 2014, ref. no. 29) stating Barley is more recalcitrant to tissue culture than the remaining cereals, however, they cited another reference in the introduction section stating that wheat is more recalcitrant one (Hiei et al. ref. 29). The authors can combine both reference in this sentence like this "Barley and wheat are more recalcitrant to tissue culture (18, 29), this makes more sense. 

Answer k: Thanks for your suggestion. The statement has been revised in the text (Line 51-53, Page 2; Line 36 in Page 11).

L-Line 1, Page 10: the genes names should be italic across the whole manuscript some were not like this one HORVU7Hr1G114450 and others, especially in first paragraph of page 10 " directly after table 1". Please, revise that across the manuscript

Answer L: Thanks for your comment. We revised it across the manuscript (Line 1-7, Page 11) and all gene names are written in italic form now.

m- Line 232, Page 13: Arabidopsis sometimes italic/regular? please, be consistent

Answer m: Sorry for that and we made the change (Line 144, Page 14; Line 174-175, Page 15).

n- Line 166, Page 14: again "the two barley not two barley"

Answer n: We added ‘the’ before ‘two barley’ in the text (Line 168, Page 15). Thanks again for all your valuable comments.

Reviewer 2 Report

This study revealed the key genes or factors controlling the callus induction and plantlet regeneration in barley, the transcriptomic profiles of immature embryos of Golden Promise and ZU9, which differed dramatically in the efficiency of the genetic transformation were detected. In total, 5,386 up-regulated and 6,257 down-regulated genes were identified in Golden Promise. Several genes, identified exclusively in GP callus, were selected for further investigation. These genes were mainly involved in protein metabolism, energy metabolism, stress response, detoxification and ubiquitin-proteasome. Four YUC, one PIN, one TAR, three 18 SAUR, three IAA and one IPT, seven CKX, three AHK, three AHP, one ARR were differentially expressed in the calli of the two barley genotypes, suggesting that biosynthesis, response and transport of auxin and cytokinin might be associated with cell reprogramming during callus induction.

Comment 1: Section introduction

Please mention the auxin signaling pathway that controls the callus induction in the introduction.

Comment 2:

Please write more details about the aim of the work.

Comment 3:

Please. Adjust the figure legends to be more clearly.

Comment 4:

English language should be improved
